# Assessing the quality and reliability of YouTube videos as a source of information on inflammatory back pain

Mete Kara[1], Erkan Ozduran[2], Müge Mercan Kara[3], Volkan Hanci[4] and Yüksel Erkin[5]

[1] Department of Rheumatology, University of Health Sciences Izmir Bozyaka Education and Research Hospital, Izmir, Turkey
[2] Department of Physical Medicine and Rehabilitation, Department of Pain Medicine, Sivas Numune Hospital, Sivas, Turkey
[3] Department of Neurology, Department of Pain Medicine, Dokuz Eylul University, Izmir, Turkey
[4] Department of Anesthesiology and Reanimation, Department of Critical Care Medicine, Dokuz Eylul University, Izmir, Turkey
[5] Department of Anesthesiology and Reanimation, Department of Pain Medicine, Dokuz Eylul University, Izmir, Turkey

Corresponding author
Erkan Ozduran,
erkanozduran@gmail.com

## ABSTRACT

**Background.** Inflammatory back pain is a chronic condition with localized pain, particularly in the axial spine and sacroiliac joints, that is associated with morning stiffness and improves with exercise. YouTube is the second most frequently used social media platform for accessing health information. This study sought to investigate the quality and reliability of YouTube videos on inflammatory back pain (IBP).

**Methods.** The study design was planned as cross-sectional. A search was conducted using the term "inflammatory back pain," and the first 100 videos that met the inclusion criteria were selected on October 19, 2023. The data of the videos selected according to the inclusion and exclusion criteria in the study settings were examined. Videos with English language, with audiovisual content, had a duration >30 s, non-duplicated and primary content related to IBP were included in the study. A number of video parameters such as the number of likes, number of views, duration, and content categories were assessed. The videos were assessed for reliability using the Journal of the American Medical Association (JAMA) Benchmark criteria and the DISCERN tool. Quality was assessed using the Global Quality Score (GQS). Continuous variables were checked for normality of distribution using Shapiro–Wilk test and Kolmogorov–Smirnov test. Kruskal–Wallis test and Mann–Whitney U test were used to analyze the continuous data depending on the number of groups. Categorical data were analyzed using Pearson's chi-square test.

**Results.** Reliability assessment based on JAMA scores showed 21% of the videos to have high reliability. Quality assessment based on GQS results showed 19% of the videos to have high quality. JAMA, DISCERN, and GQS scores differed significantly by source of video ($p < 0.001$, $< 0.001$, and $= 0.002$, respectively). Video duration had a moderate positive correlation with scores from the GQS ($r = 0.418$, $p < 0.001$), JAMA ($r = 0.484$, $p < 0.001$), and modified DISCERN ($r = 0.418$, $p < 0.001$).

**Conclusion.** The results of the present study showed that YouTube offers videos of low reliability and low quality on inflammatory back pain. Health authorities have a
responsibility to protect public health and should take proactive steps regarding health information shared on social media platforms.

## INTRODUCTION

Back pain (BP) affects millions of individuals annually and is the most common musculoskeletal complaint. It is a highly prevalent health problem that causes activity restriction and absenteeism in much of the world (*Gibbs et al., 2023*). According to a 2009–2010 report from the National Health and Nutrition Examination Survey, 20% of American individuals aged 20–70 years have chronic BP. Leading causes of chronic low BP include disc herniation, musculoskeletal strain, and osteoporotic fractures (*Rianon et al., 2022*). Differential diagnosis of BP is multifactorial, and many diseases present with similar signs or symptoms (*Gibbs et al., 2023*). Inflammatory back pain (IBP) and mechanical BP, the two most common causes of BP, can be differentiated in that IBP improves with activity, worsens with rest, is of insidious onset, persists for >3 months, and is characterized by morning stiffness (*Lassiter & Allam, 2022*).

IBP is a condition characterized by pain localized to the axial spine and sacroiliac joints. Although IBP symptoms are classically associated with ankylosing spondylitis, they can also be seen in other seronegative spondyloarthropathies such as enteropathic arthropathy, reactive arthritis, and psoriatic arthritis (*Weisman, 2012*). The prevalence of IBP was 5%–6% in individuals aged 20–69 years in the United States of America (USA) and 3%–7% in the United Kingdom (UK). In a study conducted in Mexico, the prevalence was found to be 3%. The prevalence was found to be higher in non-Hispanic white men than in non-Hispanic black men (*Lassiter & Allam, 2022*). In addition, the prevalence of IBP did not differ significantly between age groups or between men and women (*Weisman, Witter & Reveille, 2013*). Its pathophysiology is attributed to the systemic inflammatory response to inflammatory mediators that induce proinflammatory intracellular changes in the joints of the axial skeletal system (*Zhao et al., 2012*). Some of the clinical assessment criteria include the 1977 Calin criteria, the 1984 modified New York criteria, and the 2009 Assessment of SpondyloArthritis International Society (ASAS) criteria (*Weisman, 2012*). There is also a gap between diagnoses of spondyloarthropathy and IBP. Accordingly, a study estimated that 36.46% of Asian patients with inflammatory back pain met the criteria for nr-axial SpA classification. The authors consider that perhaps IBP may be a unique condition separate from other inflammatory immune-mediated conditions (*Tam et al., 2019*).

Healthcare providers have been the primary point of reference for informing patients and managing their concerns. However, dissemination of health information on social media platforms has become more widespread over time, resulting in a need to investigate whether the information uploaded on the social media is consistent with existing scientific

evidence (*Maia et al., 2021a*; *Maia et al., 2021b*). *AlMuammar et al. (2021)* found that 92.6% of the participants in their study used the Internet to seek medical information and 42% of them accessed information from Internet sources to avoid going to the hospital. The same study also reported YouTube to be the second most frequently used social media platform for accessing health information. Another study reported that 54% of patients conducted a search on their disease before doctor visits and that misinformation has a direct impact on patient decision-making and patient–physician relationship (*Hornung et al., 2022*). Thus, providing accurate information to patients is critical. The positive effects of YouTube on medical education or patient education about specific conditions are undeniable (*Sampson et al., 2013*). However, the possibility of sharing misinformation and hidden industry influence raises some concerns (*Syed-Abdul et al., 2013*; *Freeman, 2012*). As the popularity of online platforms such as YouTube continues to grow, so do concerns over the overall quality and reliability of videos uploaded to these platforms. Previous studies have reviewed the reliability and quality of YouTube videos on topics such as low BP, lumbar disc herniation, and epidural steroid injection (*Chang & Park, 2021*; *Maia et al., 2021a*; *Maia et al., 2021b*; *Mohile et al., 2023*).

The study containing LBP-related information presented on YouTube has been shown to contain information that is not evidence-based. It has been determined that there is a tendency to prioritize information about invasive methods rather than how the LBP process is *Maia et al. (2021a)* and *Maia et al. (2021b)*. In the study on epidural steroid injection, it was reported that the reliability and quality of the content was found to be low, even in the videos uploaded by doctors and hospitals (*Chang & Park, 2021*). It was a matter of curiosity what the content, reliability and quality of YouTube videos about IBP were. However, there is limited information on YouTube content related to IBP. Conducting such a study will not only shed light on the reliability and quality of the information that patients seek about IBP on the YouTube platform, but will also direct clinicians and researchers to the issues that threaten public health by conducting studies on different conditions on social media platforms. The present study sought to assess the reliability and quality of YouTube videos on IBP and identify sources that provide more reliable information.

## MATERIALS & METHODS

### Ethical approval

This study received ethics committee approval (Dokuz Eylül University Ethics Committee; no.: 2023/33-13; date: 18.10.2023).

### Study design, setting and video selection

In this cross-sectional study the data of the first 100 videos selected according to the inclusion and exclusion criteria in the study settings were examined. On October 19, 2023, a search was conducted on YouTube (https://www.youtube.com) using the term "Inflammatory back pain". The videos were selected by two authors (M.K. and E.O.). A neutral term was used to create a large pool of videos (*Barlas et al., 2023*; *Ozduran & Büyükçoban, 2022*). Possible inconsistency in video assessment was resolved by a third author (M.M.K.) making the final decision. The authors deleted cookies and Internet search

history, signed out of their Google account, and used the Google Incognito form to search YouTube (*Maia et al., 2021a*; *Maia et al., 2021b*; *Ozduran & Büyükçoban, 2022*). Videos with English language, with audiovisual content, had a duration >30 s, non-duplicated and primary content related to IBP were included in the study (*Barlas et al., 2023*; *Chang & Park, 2021*). Videos were excluded if they were not in English language and had no audiovisual content (*Chang & Park, 2021*) or had a duration <30 s, duplicated and primary content unrelated to IBP (*Barlas et al., 2023*). Videos were sorted by "view count" based on the rationale that the most viewed videos were more likely to be accessed by users seeking information in a specific area (*Chang & Park, 2021*). After a detailed assessment based on the inclusion and exclusion criteria, the first 100 videos were included in the study, as indicated in the literature (*Basch et al., 2021*; *Manchaiah et al., 2020*). The present study follows the STROBE (Strengthening the Reporting of Observational Studies in Epidemiology) reporting guidelines (*von Elm et al., 2007*).

## Content categories

The videos were thoroughly analyzed in terms of content type relating to IBP. The type of content was divided into six groups to determine whether videos included them. These content categories were: (1) etiology, (2) symptoms, (3) physical examination, (4) diagnosis, (5) differential diagnosis, and (6) treatment.

## Reliability assessment

The videos were assessed for reliability based on "The Journal of the American Medical Association (JAMA) Benchmark." The JAMA Benchmark criteria are an objective set of guidelines used for assessing the reliability of online resources such as websites, videos, and podcasts. In this set of criteria, videos were assessed based on four categories: (1) authorship, (2) disclosure, (3) currency, and (4) attribution, where each category was assigned 1 point, yielding a final score of 0 to 4. A video with a higher JAMA score was considered to be more reliable. Thus, videos with a JAMA a score ≤2 points are considered to have low reliability and videos with ≥3 points are considered to have high reliability (*Silberg, Lundberg & Musacchio, 1997*) (Table 1). In assessing JAMA results, videos with a score of 0 and 1 are considered to contain insufficient data, videos with a score of 2 and 3 are considered to contain partially sufficient data, and videos with a score of 4 are considered to contain completely sufficient data (*Ozduran & Büyükçoban, 2022*).

The videos were also assessed for reliability using the modified DISCERN scale. The tool was created by the Public Health and Primary Care Division of Oxford University in 1999 under its original name Quality Criteria for Consumer Health Information to assess the quality of information and treatment options for health problems (*Rodriguez-Rodriguez et al., 2022*). The DISCERN tool consists of five criteria, where the video is assigned one point if it meets the relevant criterion and zero point if it does not. The final score ranges from 0 to 5, with higher scores representing higher reliability (*Chang & Park, 2021*) (Table 1). Validity and reliability evaluations were made for the JAMA and DISCERN scales (*Silberg, Lundberg & Musacchio, 1997*; *Charnock et al., 1999*).

**Table 1 Contents DISCERN, GQS and JAMA assessment criteria.**

| JAMA benchmark criteria | Total score (0–4 Points) |
| --- | --- |
| Currency | 1 point (Dates of the content should be indicated). |
| Disclosure | 1 point (Funding, Conflicts of interest, sponsorship, support, video ownership and advertising should be fully disclosed) |
| Authorship | 1 point (Contributors and Authors, relevant credentials, and affiliations, should be provided) |
| Attribution | 1 point (Sources and References should be listed) |
| **Modified DISCERN Criteria** | **Total Score (0-5 Points)** |
| 1 Are additional sources of information listed for patient reference? | 0-1 point |
| 2 Does the video address areas of contro-versy/uncertainty? | 0-1 point |
| 3 Is the video clear, concise, and understandable? | 0-1 point |
| 4 Are valid sources cited? | 0-1 point |
| 5 Is the provided information balanced and unbiased? | 0-1 point |
| **GQS** | **Score** |
| Poor quality, poor flow of the site, most information missing, not at all useful for patients | 1 |
| Generally poor quality and poor flow, some information listed but many important topics missing, of very limited use to patients | 2 |
| Moderate quality, suboptimal flow, some important information is adequately discussed but others poorly discussed, somewhat useful for patients | 3 |
| Good quality and generally good flow, most of the relevant information is listed, but some topics not covered, useful for patients | 4 |
| Excellent quality and excellent flow, very useful for patients | 5 |

**Notes.**
JAMA, Journal of the American Medical Association; GQS, Global Quality Score.

## Quality assessment

The Global Quality Score (GQS) is used to assess the quality of all resources available online. In this scoring, each criterion is worth a score of 1, and a total score of 5 indicates excellent quality. Content with a final score of 4 or 5 is considered to be of high quality, 3 indicates moderate quality, and a score of 1 and 2 indicates poor quality (*Chang & Park, 2021*) (Table 1). The GQS reveals the accessibility and quality of information, as well as potential usefulness for any user (*Rodriguez-Rodriguez et al., 2022*). Validity and reliability evaluations were made for the GQS scales (*Bernard et al., 2007*).

## Video sources

Video sources were classified into academic institutions, health-related websites, professional organizations/societies, physicians, patients, news channels, and commercial and nonprofit organizations. The presence/absence of animation content in the videos and the country and continent of origin were also recorded.

## Assessment of user engagement (video parameters)

For each video, six parameters were recorded: (1) views, (2) likes, (3) dislikes, (4) video power index (VPI), (5) video duration (s), and (6) comments. Video popularity was assessed using the video power index (VPI) ([likes count]/[dislikes count + likes count] ×100) (*Reina-Varona et al., 2022*).

## Statistical analysis

The study data were analyzed using SPSS (Statistical Package for Social Sciences, Chicago, IL, USA) 24.0 software. Continuous data were presented using means and standard deviation (mean ±standard deviation (SD)), and categorical data were presented using percentage (%) and number ($n$). Continuous variables were checked for normality of distribution using Shapiro–Wilk test and Kolmogorov–Smirnov test. Kruskal–Wallis test and Mann–Whitney $U$ test were used to analyze the continuous data depending on the number of groups. Categorical data were analyzed using Pearson's chi-square test. Groups were compared for correlation analysis using Pearson's correlation test. Statistical significance was set at $p < 0.05$.

## RESULTS

YouTube was searched using a search term, and the first 100 videos with the highest count of views that met the inclusion criteria were included in the study. The videos were thoroughly reviewed based on the exclusion criteria, resulting in the exclusion of 10 videos in non-English language, five irrelevant videos, one repeated video, eight videos under 30 s and 12 videos with inadequate audiovisual content (Fig. 1). Of the repeated videos, only one was included in assessment. The total duration of the included video content was 21 h, 45 min, and 50 s. The shortest and the longest video had a duration of 36 s and 1 h, 27 min, and 26 s, respectively. The least and the most viewed videos had 53 views and 11 million views, respectively. The video with the lowest and highest number of likes had zero likes and 133,000 likes, respectively. The minimum and maximum number of comments were zero and 4,720, respectively. Only 29% ($n = 29$) of the videos had animation content. The mean count of likes and dislikes for all the videos were $4,672.12 \pm 18,725.93$ and $115.32 \pm 450.82$, respectively. Mean views, comments, duration, and VPI were $287,700.93 \pm 1,253, 532.21$, $167.29 \pm 595.21$, $783.5 \pm 1,063.36$, and $96.87 \pm 5.22$, respectively. The time period from 2020 onwards was the period with the highest number of uploads (54% of all videos). Analysis of the sources showed that the top two sources of upload were health-related websites ($n = 31$) and professional organizations/societies ($n = 21$) (Fig. 2).

As for the analysis of content categories, the two most common topics covered in the videos were symptoms (75%) and treatment (59%). Videos with diagnosis-related content were uploaded in significantly higher numbers from 2020 onwards ($p = 0.037$). Analysis of other content categories and animation content did not show any significant difference by year (Table 2).

In reliability and quality assessment, there is statistically significant and significant agreement between raters in the JAMA, DISCERN and GQS evaluations, respectively ($\kappa = 0.94$, $p < 0.001$; $\kappa = 0.96$, $p < 0.001$; $\kappa = 0.94$, $p < 0.001$). Reliability was assessed

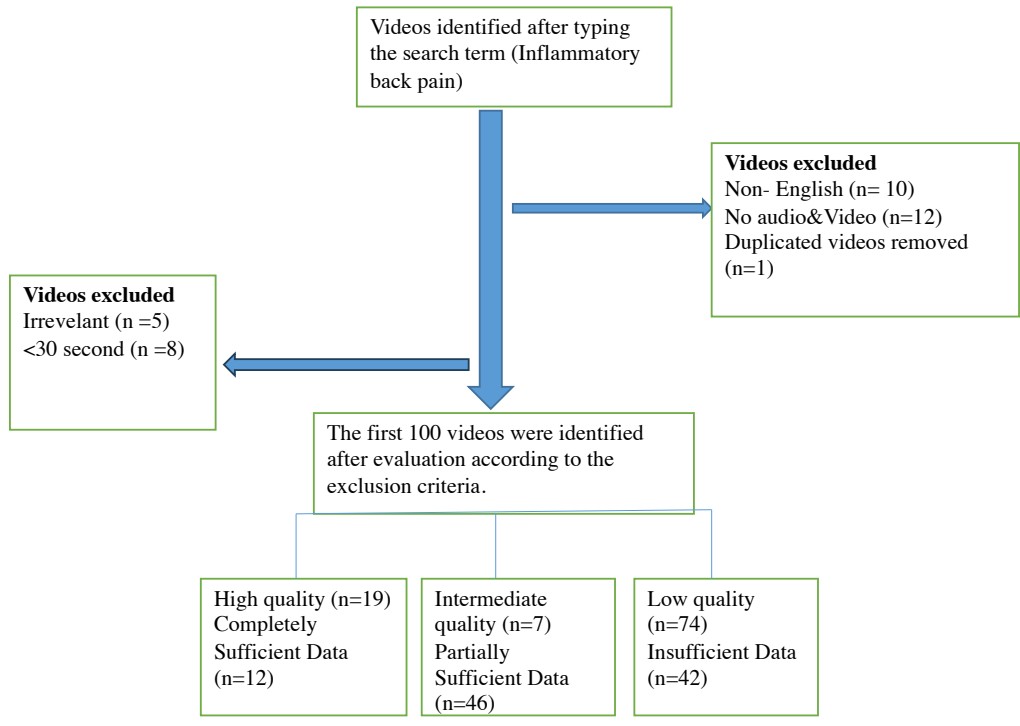

**Figure 1** Flow diagram of YouTube videos on inflammatory back pain.

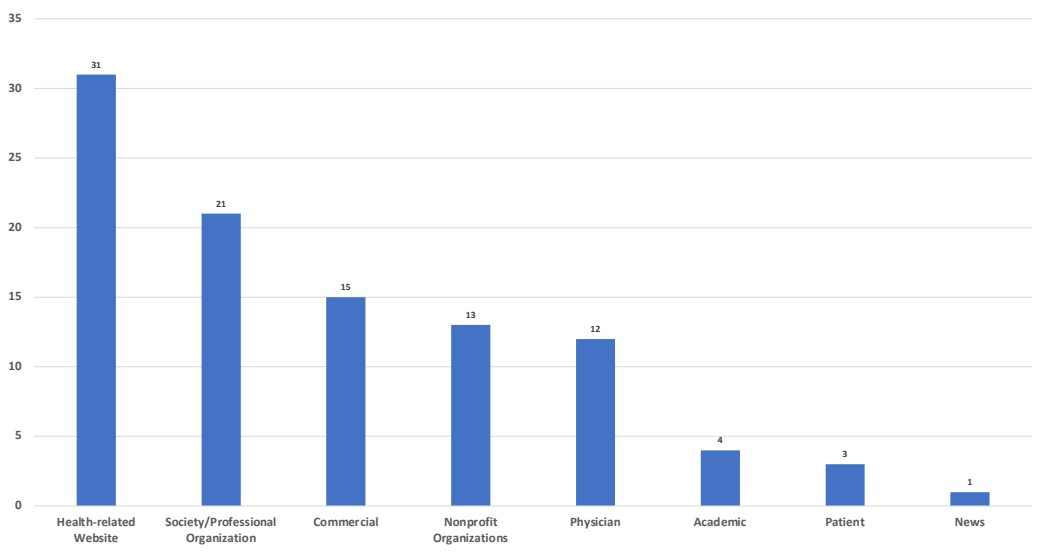

**Figure 2** Sources of videos.

using the JAMA and modified DISCERN tools. Mean ± SD values of JAMA and modified DISCERN scores were 1.88 ± 1.04 and 37.1 ± 20.37, respectively. When JAMA scores were categorized into high and poor reliability, 21 (21%) and 79 (79%) videos were of high and

**Table 2 Comparison of the quality, content and quality of videos over the years.**

| Video content/years | | 2023–2020, n (%) | 2019–2015, n (%) | 2014–2011, n (%) | p |
|---|---|---|---|---|---|
| Animation | + | 16(55.2%) | 13(44.8%) | 0(0%) | 0.060 |
| | - | 38(53.5%) | 22(31%) | 11(15.5%) | |
| Etiology | + | 24(57.1%) | 13(31%) | 5(11.9%) | 0.769 |
| | – | 30(51.7%) | 22(37.9%) | 6(10.3%) | |
| Symptoms | + | 42(56%) | 25(33.3%) | 8(10.7%) | 0.235 |
| | – | 12(48%) | 10(40%) | 3(12%) | |
| Physical Examination | + | 24(55.8%) | 14(32.6%) | 5(11.6%) | 0.904 |
| | – | 30(52.6%) | 21(36.8%) | 6(10.5%) | |
| Diagnosis | + | 36(65.5%) | 15(27.3%) | 4(7.3%) | **0.037** |
| | – | 18(40%) | 20(44.4%) | 7(15.6%) | |
| Differential Diagnosis | + | 24(47.1%) | 22(43.1%) | 5(9.8%) | 0.219 |
| | – | 30(61.2%) | 13(26.5%) | 6(12.2%) | |
| Treatment | + | 36(61%) | 20(33.9%) | 3(5.1%) | 0.051 |
| | – | 18(43.9%) | 15(36.6%) | 8(19.5%) | |
| JAMA | Insufficient data (1 Point) | 27(64.3%) | 11(26.2%) | 4(9.5%) | 0.402 |
| | Partially sufficient data (2 or 3 points) | 20(43.5%) | 20(43.5%) | 6(13%) | |
| | Completely sufficient data (4 points) | 7(58.3%) | 4(33.3%) | 1(8.3%) | |
| JAMA | Low Reliable | 43(54.4%) | 26(32.9%) | 10(12.7) | 0.491 |
| | High Reliable | 11(52.4%) | 9(42.9%) | 1(4.8%) | |
| GQS | Low quality (1 or 2 points) | 42(56.8%) | 22(29.7%) | 10(13.5%) | **0.046** |
| | Intermediate quality (3 points) | 1(14.3%) | 6(85.7%) | 0(0%) | |
| | High quality (4-5 points) | 11(57.9%) | 7(36.8%) | 1(5.3%) | |
| Modified DISCERN | Very Poor | 21(65.6%) | 9(28.1%) | 2(6.3%) | 0.288 |
| | Poor | 16(50%) | 9(28.1%) | 7(21.9%) | |
| | Fair | 7(41.2%) | 9(52.9%) | 1(5.9%) | |
| | Good | 5(55.6%) | 4(44.4%) | 0(0%) | |
| | Excellent | 5(50%) | 4(40%) | 1(10%) | |

**Notes.**
Pearson Chi-square test.
JAMA, Journal of the American Medical Association benchmark criteria; GQS, Global Quality Score.
Bold font indicates statistical significance.

poor reliability, respectively. Assessment based on the DISCERN tool showed ten (10%) videos to be excellent and nine (9%) videos to be good. Quality assessment using the GQS yielded a mean $\pm$ SD score of 2.13 $\pm$ 1.2, and only 19 (19%) of the videos were of high quality (Table 3).

Reliability and quality assessment results differed significantly by video source. JAMA, DISCERN, and GQS results exhibited significant differences in connection with video source ($p < 0.001$, $<0.001$, and $= 0.002$, respectively). Of the videos considered completely sufficient according to JAMA results, 33.3% ($n = 4$) had been uploaded by academic institutions and another 33.3% ($n = 4$) by professional organizations/societies. Of the videos with insufficient data, 36.5% ($n = 27$) had been uploaded by health-related websites and 20.3% ($n = 15$) by commercial sources. Of the videos found to have good and

Kara et al. (2024), *PeerJ*, DOI 10.7717/peerj.17215

**Table 3** **Video sources by quality and reliability parameters.**

| | | Academic (n) | Physician (n) | Society/ Professional organization (n) | Health-related Website (n) | Patient (n) | News (n) | Commercial (n) | Nonprofit organization | *p* |
|---|---|---|---|---|---|---|---|---|---|---|
| GQS(1-5 points) | Low quality (1 or 2 points) (*n* = 74) | 0(0%) | 9(12.2%) | 11(14.9%) | 27(36.5%) | 3(4.1%) | 1(1.4%) | 15(20.3%) | 8(10.8%) | |
| | Intermediate quality (3 points) (*n* = 7) | 0(0%) | 1(14.3%) | 3(42.9%) | 1(14.3%) | 0(0%) | 0(0%) | 0(0%) | 2(28.6%) | **0.002** |
| | High quality (4-5 points) (*n* = 19) | 4(21.1%) | 2(10.5%) | 7(36.8%) | 3(15.8%) | 0(0%) | 0(0%) | 0(0%) | 3(15.8%) | |
| JAMA score (0-4 Points) | Insufficient data (1 Point) (*n* = 42) | 0(0%) | 2(4.8%) | 3(7.1%) | 12(28.6%) | 3(7.1%) | 1(2.4%) | 15(35.7%) | 6(14.3%) | |
| | Partially sufficient data (2 or 3 points) (*n* = 46) | 0(0%) | 8(17.4%) | 14(30.4%) | 19(41.3%) | 0(0%) | 0(0%) | 0(0%) | 5(10.9%) | **<0.001** |
| | Completely sufficient data (4 points) (*n* = 12) | 4(33.3%) | 2(16.7%) | 4(33.3%) | 0(0%) | 0(0%) | 0(0%) | 0(0%) | 2 (16.7%) | |
| Modified DISCERN score (0-5 points) | Very Poor (*n* = 32) | 0(0%) | 1(3.1%) | 2(6.3%) | 7(21.9%) | 3(9.4%) | 1(3.1%) | 15(46.9%) | 3(9.4%) | |
| | Poor (*n* = 32) | 0(0%) | 3(9.4%) | 7(21.9%) | 15(46.9%) | 0(0%) | 0(0%) | 0(0%) | 7(21.9%) | |
| | Fair (*n* = 17) | 0(0%) | 6(35.3%) | 3(17.6%) | 7(41.2%) | 0(0%) | 0(0%) | 0(0%) | 1(5%) | **<0.001** |
| | Good (*n* = 9) | 0(0%) | 0(0%) | 7(77.8%) | 2(22.2%) | 0(0%) | 0(0%) | 0(0%) | 0(0%) | |
| | Excellent (*n* = 10) | 4(40%) | 2(20%) | 2(20%) | 0(0%) | 0(0%) | 0(0%) | 0(0%) | 2(20%) | |

**Notes.**

Pearson Chi-square test.

GQS, Global Quality Score; JAMA, Journal of the American Medical Association benchmark criteria.

Bold font indicates statistical significance.

excellent reliability according to the modified DISCERN score, 47.4% ($n = 9$) had been uploaded by professional organizations/societies and 21.1% ($n = 4$) by academics. As for videos considered to be of high quality according to the GQS results, 36.8% ($n = 7$) had been uploaded by professional organizations/societies and 21.1% ($n = 4$) by academics. Of the low quality videos, on the other hand, 36.5% had been uploaded by health-related websites and 20.3% ($n = 15$) by commercial sources. According to the JAMA, modified DISCERN, and GQS scores, all videos uploaded by academics were of high quality, contained completely sufficient data, and had excellent reliability. In contrast, all commercial videos were of low quality, contained insufficient data, and had very poor reliability (Table 3).

JAMA reliability scores were compared with video parameters, and only duration was associated with a significant difference ($p < 0.001$). Similarly, comparison of the modified DISCERN scores and video parameters showed only duration to be associated with a significant difference ($p < 0.001$). Videos with short duration had lower reliability scores in both the modified DISCERN and JAMA scales. Assessment of quality scores (GQS) by video parameter showed statistically significant differences in views ($p = 0.022$), dislikes ($p = 0.044$), duration ($p = 0.004$), and VPI ($p < 0.001$) (Table 4).

Video duration had a moderate positive correlation with the GQS ($r = 0.418$, $p < 0.001$), JAMA ($r = 0.484$, $p < 0.001$), and modified DISCERN ($r = 0.418$, $p < 0.001$). Thus, videos with higher quality and reliability had longer durations on average. The JAMA score had a very strong positive correlation with both the modified DISCERN ($r = 0.905$, $p < 0.001$) and GQS ($r = 0.935$, $p < 0.001$) scores (Table 5).

Comparison between the countries of origin of videos and video parameters revealed a significant difference in views ($p = 0.008$), likes ($p = 0.013$), dislikes ($p = 0.011$), and VPI ($p = 0.041$). Videos that originated from Canada had higher values for view count, likes, and dislikes than videos uploaded from other countries. Videos originating from the UK, on the other hand, had significantly higher VPI values. Assessment of the videos by continent of origin showed that videos originating from the American continent differed significantly in terms of views ($p = 0.021$), likes ($p = 0.046$), and comments ($p = 0.037$) compared with those from non-American continent videos (Table 6).

## DISCUSSION

In the present study, YouTube videos on IBP were assessed in terms of user engagement criteria, reliability, content categories, and quality. The study focused on YouTube content on IBP, a symptom that may cause life-threatening health problems and negatively affect public health in case of delayed diagnosis. Chronic BP is a common symptom leading to deterioration of health and use of health resources in the community. Four out of five individuals are known to suffer from back pain at one point in their lives. In some, these symptoms resolve over time, while in others, they become chronic (*Nieminen, Pyysalo & Kankaanpää, 2021*). Axial spondyloarthritis (axSpA) group of diseases presenting with chronic IBP lack precise symptoms at the time of initial presentation, have a slow or delayed progression, lack reliable diagnostic tests, and have low prevalence in the

**Table 4** **Video parameters according to years, and quality and reliability parameters (mean ±standard deviation).**

| Years | View Mean ± SD | Like Mean ± SD | Dislike Mean ± SD | Comment Mean ± SD | Duration Mean ± SD | VPI Mean ± SD |
|---|---|---|---|---|---|---|
| 2020-2023($n = 54$) | 245527.46 ± 837157.79 | 5766.48 ± 21386.34 | 117.02 ± 391.86 | 245.65 ± 776.54 | 1062.17 ± 1272.7 | 98.65 ± 2.51 |
| 2015-2019 ($n = 35$) | 431057.43 ± 1853983.19 | 4374.37 ± 17310.17 | 145.51 ± 589.75 | 93.03 ± 261.96 | 476.74 ± 634.13 | 95.89 ± 4.4 |
| 2011-2014 ($n = 11$) | 38600 ± 73673.27 | 247.19 ± 591.37 | 10.9 ± 22.74 | 18.9 ± 36.92 | 391.55 ± 569.83 | 91.25 ± 10.75 |
| **p** | **0.022** | 0.116 | **0.044** | 0.541 | **0.004** | **0.001>** |
| **Video Sources** | | | | | | |
| Academic($n = 4$) | 41829.5 ± 54888.59 | 274.25 ± 358.93 | 19.5 ± 33.07 | 25.25 ± 37.21 | 1690.25 ± 1215.48 | 96.35 ± 3.36 |
| Physician($n = 12$) | 244903.08 ± 801381.09 | 7416 ± 22024.46 | 169.33 ± 486.52 | 249.25 ± 630.62 | 579.58 ± 380.37 | 97.72 ± 3.33 |
| Society/Professional Organization($n = 21$) | 609228.91 ± 2387771.78 | 5587.19 ± 21966.86 | 194.19 ± 760.05 | 114.9 ± 319.57 | 1281.14 ± 1616.46 | 96.27 ± 4.92 |
| Health-related Website($n = 31$) | 128144.25 ± 326314.18 | 2284.61 ± 6797.35 | 67.52 ± 221.23 | 66.45 ± 211.85 | 509.45 ± 854.31 | 96.41 ± 6.18 |
| Patient($n = 3$) | 1735.33 ± 2556.3 | 61.67 ± 87.21 | 0.66 ± 1.15 | 15.33 ± 26.55 | 1351.67 ± 437.36 | 99.59 ± 0.7 |
| News($n = 1$) | 2022 ± 0 | 278 ± 0 | 21 ± 0 | 83 ± 0 | 109 ± 0 | 92.98 ± 0 |
| Commercial(15) | 536743.6 ± 1358575.22 | 12460.47 ± 34555.81 | 211.4 ± 525.85 | 585.6 ± 1302.75 | 379.73 ± 274.73 | 98.54 ± 1.91 |
| Nonprofit Organizations($n = 13$) | 18405.92 ± 44003.39 | 123 ± 242.6 | 4.38 ± 9.63 | 19.31 ± 45.34 | 929 ± 1113.02 | 96.04 ± 7.88 |
| **p** | 0.230 | 0.399 | 0.502 | 0.315 | 0.068 | 0.500 |
| **GQS (1-5 points)** | | | | | | |
| Low quality (1 or 2 points) ($n = 74$) | 360576.31 ± 1448599.1 | 6010.7 ± 21602.52 | 144.85 ± 520.08 | 210.53 ± 685.67 | 470.32 ± 513.88 | 96.57 ± 5.68 |
| Intermediate quality (3 points) ($n = 7$) | 262142.86 ± 298278.63 | 2867.43 ± 4066.35 | 100 ± 113.35 | 131.14 ± 151.1 | 217.43 ± 155.95 | 95.12 ± 5 |
| High quality (4-5 points) ($n = 19$) | 13286.63 ± 28606.23 | 123.58 ± 217.75 | 5.95 ± 16.45 | 12.21 ± 25.05 | 2211.8 ± 1568.1 | 98.7 ± 2.4 |
| **p** | **0.003** | **0.007** | **0.006** | **0.013** | **<0.001** | **0.038** |
| **JAMA score (0-4 Points)** | | | | | | |
| Insufficient data (1 Point) ($n = 42$) | 273883.62 ± 863588.46 | 6016.83 ± 21563.1 | 121.02 ± 366.97 | 262.31 ± 819.96 | 436.71 ± 456.46 | 96.93 ± 5.13 |
| Partially sufficient data (2 or 3 points) ($n = 46$) | 370230.33 ± 1658574.93 | 4618.70 ± 18458.18 | 137.76 ± 566.39 | 119.87 ± 387.05 | 712.61 ± 1074.87 | 96.51 ± 5.79 |
| Completely sufficient data (4 points) ($n = 12$) | 19698.83 ± 34844.40 | 170.42 ± 263.15 | 9.33 ± 20.22 | 16.50 ± 30.73 | 2269 ± 1336.46 | 97.98 ± 2.79 |
| **p** | 0.452 | 0.531 | 0.434 | 0.509 | **<0.001** | 0.827 |
| **Modified DISCERN score (0-5 points)** | | | | | | |
| Very Poor ($n = 32$) | 314921.47 ± 972410.46 | 7049.78 ± 24428.72 | 141.62 ± 415.25 | 327.72 ± 931.59 | 380.25 ± 414.94 | 97.78 ± 3.39 |
| Poor ($n = 32$) | 83865.65 ± 200762.17 | 1218.78 ± 3893.48 | 29.53 ± 67.73 | 42.63 ± 75.32 | 480.56 ± 457.59 | 94.83 ± 7.53 |
| Fair ($n = 17$) | 860213.82 ± 2698727.8 | 11129 ± 29667.82 | 324.59 ± 915.34 | 250.35 ± 619.55 | 724.88 ± 655.69 | 97.92 ± 2.83 |
| Good ($n = 9$) | 129018.66 ± 274248.34 | 1279.78 ± 3288.63 | 48.11 ± 94 | 47.89 ± 83.68 | 2332.33 ± 2049.46 | 97.45 ± 4.87 |
| Excellent ($n = 10$) | 22410.2 ± 37779.68 | 190.7 ± 284.58 | 10.4 ± 22.1 | 19.1 ± 33.28 | 1979.4 ± 1246.4 | 98.15 ± 2.76 |
| **p** | 0.705 | 0.796 | *0.569* | 0.826 | **<0.001** | 0.417 |

**Notes.**

Kruskal Wallis Test

n, Number of videos; SD, Standart Deviation; GQS, Global Quality Score; JAMA, Journal of the American Medical Association benchmark criteria; VPI, Video Power Index.

Bold font indicates statistical significance ($p < 0.05$).
**Table 5  Correlation analysis.**

| | GQS | | JAMA | | Modified DISCERN | | Number of views | | Number of likes | | Number of dislikes | | Number of comments | | Video duration | |
|---|---|---|---|---|---|---|---|---|---|---|---|---|---|---|---|---|
| | r | p | r | p | r | p | r | p | r | p | r | p | r | p | r | p |
| JAMA | 0.935* | <**0.001** | | | | | | | | | | | | | | |
| Modified DISCERN | 0.897* | <**0.001** | 0.905* | <**0.001** | | | | | | | | | | | | |
| Number of views | −0.093 | 0.356 | −0.051 | 0.617 | −0.037 | 0.713 | | | | | | | | | | |
| Number of likes | −0.075 | 0.461 | 0.032 | 0.752 | −0.011 | 0.910 | 0.924* | <**0.001** | | | | | | | | |
| Number of dislikes | −0.144 | 0.153 | −0.100 | 0.321 | −0.093 | 0.360 | 0.867* | <**0.001** | 0.879* | <**0.001** | | | | | | |
| Number of comments | −0.111 | 0.270 | −0.080 | 0.428 | −0.079 | 0.437 | 0.750* | <**0.001** | 0.772* | <**0.001** | 0.730* | <**0.001** | | | | |
| Video duration;second | 0.418* | <**0.001** | 0.484* | <**0.001** | 0.418* | <**0.001** | −0.140 | 0.165 | −0.047 | 0.640 | −0.116 | 0.252 | −0.008 | 0.938 | | |
| VPI | 0.072 | 0.479 | 0.053 | 0.598 | 0.056 | 0.582 | −0.616* | <**0.001** | −0.582* | <**0.001** | −0.811* | <**0.001** | −0.470* | <**0.001** | 0.275* | **0.006** |

**Notes.**

Spearman's correlation test.

GQS, Global Quality Score; JAMA, Journal of the American Medical Association benchmark criteria; VPI, Video Power Index.

Bold font indicates statistical significance.

Kara et al. (2024), PeerJ, DOI 10.7717/peerj.17215

**Table 6  Video parameters by continent and country.**

| | View Mean ± SD | Like Mean ± SD | Dislike Mean ± SD | Comment Mean ± SD | Duration Mean ± SD | VPI Mean ± SD | Animation n(%) | JAMA Mean ± SD | DISCERN Mean ± SD | GQS Mean ± SD |
|---|---|---|---|---|---|---|---|---|---|---|
| **Continent** | | | | | | | | | | |
| America (n = 69) | 375761.04 ± 1494647.75 | 6082.84 ± 22223.85 | 152.84 ± 535.61 | 194.22 ± 660.30 | 763.1 ± 1060.1 | 96.47 ± 5.18 | 19(65.5%) | 1.96 ± 1 | 39.25 ± 20.8 | 2.16 ± 1.2 |
| Non-America (n = 31) | 91696.16 ± 253234.97 | 1532.13 ± 4786.18 | 31.81 ± 100.01 | 107.35 ± 418.89 | 828.9 ± 1086.76 | 97.75 ± 5.29 | 10(34.5%) | 1.71 ± 1.13 | 32.32 ± 18.83 | 2.06 ± 1.2 |
| P | **0.021** | **0.046** | 0.058 | **0.037** | 0.929 | 0.051 | 0.399 | 0.149 | 0.096 | 0.625 |
| **Country** | | | | | | | | | | |
| USA (n = 64) | 184603.92 ± 696725.63 | 3715.86 ± 17419.12 | 81.14 ± 296.71 | 148.52 ± 614.19 | 775.56 ± 1096.47 | 96.43 ± 5.34 | 16(55.2%) | 1.95 ±.1.01 | 38.84 ± 21.32 | 2.16 ± 1.22 |
| UK (n = 10) | 206878.2 ± 418982.32 | 3875.7 ± 8097.11 | 80.7 ± 169.37 | 236.7 ± 728.97 | 1209.5 ± 1524.2 | 99.39 ± 093 | 4(13.8%) | 1.6 ± 1.35 | 30.2 ± 22.87 | 1.7 ± 1.25 |
| Australia (n = 10) | 40695.9 ± 101201.76 | 306.7 ± 709.22 | 7.5 ± 13.6 | 21.7 ± 38.81 | 711 ± 1025.69 | 94.48 ± 8.6 | 1(3.4%) | 1.7 ± 0.94 | 31.8 ± 12.52 | 2.2 ± 0.91 |
| Canada (n = 4) | 3528000 ± 5134932.46 | 45475 ± 51208.16 | 1338.25 ± 1631.79 | 974 ± 1045.39 | 604.5 ± 424.54 | 96.35 ± 2.04 | 3(10.3%) | 1.75 ± 0.5 | 40 ± 9.23 | 1.75 ± 0.5 |
| Others (n = 12) | 30641.75 ± 73606.05 | 472.75 ± 1025.5 | 8.67 ± 21.27 | 62 ± 139.5 | 590.92 ± 508.02 | 99.29 ± 0.9 | 5(17.2%) | 1.91 ± 1.16 | 37 ± 21.5 | 2.42 ± 1.44 |
| P | **0.008** | **0.013** | **0.011** | 0.100 | 0.629 | **0.041** | 0.096 | 0.626 | 0.512 | 0.482 |

**Notes.**

Mann Whitney U test in analysis of continents, Kruskal Wallis Test test in analysis of countries.

USA, United States of America; VPI, Video Power Index; SD, Standart Deviation; GQS, Global Quality Score; JAMA, Journal of the American Medical Association benchmark criteria.

Bold font indicates statistical significance.
community, which may delay diagnosis by up to 8–10 years (*Magrey et al., 2020*; *Sykes et al., 2015*). Delayed diagnosis may lead to deterioration in patient's quality of life and increased economic burden, as well as severe disability due to untreated disease (*Vangeli et al., 2015*; *Juanola Roura et al., 2015*). The ASAS criteria have guided the diagnosis of axSpA by detecting the different signs and symptoms, including IBP, that manifest in early stages of the disease. Further, general practitioners, usually the first point of contact for patients, have good knowledge of IBP symptoms but not of disease-specific features and exhibit modest confidence in evaluating patients with IBP, which may cause delays in the diagnosis of the disease (*Aljohani, Barradah & Kashkari, 2022*). A number of factors including chronicization of symptoms and lack of response to treatment prompt patients to turn to popular social media platforms to seek information on their disease. YouTube, a platform with a billion views per month, has become a medium that influences individuals' decision-making about their health (*Maia et al., 2021a*; *Maia et al., 2021b*). The present study sought to investigate whether YouTube provides reliable, high-quality, and accurate information to individuals searching that platform for information on IBP. The results of the present study showed that the number of videos on IBP uploaded on YouTube from 2020 onwards has increased more than in any other time period, and diagnosis-related content has become a more common topic in videos uploaded in recent years. The study also showed that the highest proportion of the videos were uploaded by health-related websites (31%), and that the JAMA, DISCERN, and GQS scores were higher for videos uploaded by professional organizations/societies and academics but lower for videos uploaded by health-related websites and commercial sources. Videos with high JAMA and DISCERN scores had longer durations.

The top two sources of upload for the videos investigated in this study were health-related websites (31%) and professional organizations/societies (21%). This result is consistent with some previous studies that found health-related websites to be the leading source of videos on various topics (*Duman, 2020*; *Onder, Onder & Zengin, 2022*). YouTube channels managed by health-related websites are often created by nonphysicians and are frequently used to upload videos on popular topics.

Analysis of content categories in the present study showed that the most frequently mentioned topic in the videos was IBP symptoms, followed by treatment-related content. This is consistent with previous studies that found symptom content to be the most common topic in videos (*Tang, Olscamp & Choi SK, 2017*; *Ozsoy-Unubol & Alanbay-Yagci, 2021*). For instance, *Ozsoy-Unubol & Alanbay-Yagci (2021)* investigated YouTube videos on fibromyalgia, another rheumatic disease, and emphasized that symptom- and treatment-related content were the most common categories mentioned in videos. This suggests that in videos on rheumatic diseases, disease symptoms and treatment-related issues attract viewers' attention the most, which prompts video uploaders to focus on these topics. Another interesting finding in our study is the increase in diagnosis-related content in videos uploaded in recent years. This can be explained by the increasing awareness about the loss of health caused by delayed diagnosis in people with IBP symptoms, resulting in diagnosis-related content being more frequently mentioned in recent videos.

*Lombo-Moreno et al. (2023)* found high DISCERN scores in videos uploaded by professional organizations. *Aglamis, Senel & Koudonas (2023)* found high GQS and DISCERN scores for videos uploaded by professional organizations and academic institutions. *Wu et al. (2022)* reported high JAMA and DISCERN scores in videos from professional organizations and universities. Our study found similar results in that it observed high JAMA, DISCERN, and GQS scores in videos from professional organizations and academic sources. In contrast, reliability and quality scores were low in videos uploaded by commercial and health-related websites. This study found health-related websites to be the leading source of videos, and this result indicates a need for videos containing reliable and quality information to protect public health and provide accurate information for viewers. Thus, YouTube should prevent misinformation by establishing rigorous control mechanisms to filter out videos of low quality and poor reliability.

We found video duration to have a positive moderate correlation with the JAMA, DISCERN, and GQS scores. The literature emphasizes that videos with high reliability and quality scores have longer durations (*Kyarunts et al., 2022*). Viewers seeking accurate and quality information should thus be skeptical of short videos. The present study found, as in previous studies, that the JAMA, DISCERN, and GQS scores had a moderate correlation with one another (*Bolac, Ozturk & Yildiz, 2022*). This can be explained by the fact that quality videos are reliable and reliable videos offer quality information.

Breakdown of the countries and continents of origin in the present study showed that most of the videos were uploaded by YouTube channels originating from the USA (64%) and the American continent (69%). Similarly, previous studies reported that most of the videos originated from the USA (*Li, Giuliani & Ingledew, 2021*). Results also showed that videos originating from the American continent had a significant association with the view count, likes, and comments. This can be explained by the fact that YouTube, an organization originating in the USA, is actively used by users from the USA to provide information to the whole world, and this translates into engagement in terms of certain video parameters. Thus, YouTube seems to assume an important function by conveying health information and has a direct impact on public health.

## Limitations

This study has some limitations: the exclusion of videos in non-English languages, the possibility of the cross-sectional study to yield different results in another time period, and the part of subjective assessment by the authors despite the use of objective questionnaires. Although the search in this study was conducted using the Google Incognito form, YouTube's unique video ranking feature can be considered as another limitation.

## Strengths of this study

Although there are studies on mechanical back pain on YouTube, there are no YouTube studies on inflammatory back pain. Therefore, it can be stated that our study will be a guide for patients who are looking for content on YouTube regarding inflammatory back pain. In addition, studies on social media are of great importance in terms of protecting public health. It can be stated that with developing technology, such studies will be needed

more in the future to ensure that individuals have access to accurate, reliable and quality information.

## CONCLUSION

Analysis of YouTube videos on IBP in the present study showed the majority of the videos were of low quality and reliability. Videos uploaded by professional organizations/societies and academics contained more reliable and higher-quality information. It is likely that that digitalization will make YouTube more prominent in the field of health, and viewers should be encouraged to exercise caution when approaching information received from that platform. Health authorities should be encouraged to work on social media practices involving sharing health information to protect public health. It is clear that new studies on different topics to be conducted on social media platforms in the future will help raise patient awareness and support public health.

### Funding
The authors received no funding for this work.

### Competing Interests
The authors declare there are no competing interests.

### Author Contributions
- Mete Kara conceived and designed the experiments, performed the experiments, prepared figures and/or tables, authored or reviewed drafts of the article, and approved the final draft.
- Erkan Ozduran conceived and designed the experiments, performed the experiments, analyzed the data, prepared figures and/or tables, authored or reviewed drafts of the article, and approved the final draft.
- Müge Mercan Kara conceived and designed the experiments, authored or reviewed drafts of the article, and approved the final draft.
- Volkan Hanci conceived and designed the experiments, analyzed the data, authored or reviewed drafts of the article, and approved the final draft.
- Yüksel Erkin conceived and designed the experiments, analyzed the data, authored or reviewed drafts of the article, and approved the final draft.

### Ethics
The following information was supplied relating to ethical approvals (*i.e.,* approving body and any reference numbers):

The University of Dokuz Eylül granted Ethical approval to carry out the study within its facilities (Ethical Application Ref: 2023/33-13, Date:18.10.2023).

## Data Availability

The raw data is available in the Supplemental File.

## Supplemental Information

Supplemental information for this article can be found online at http://dx.doi.org/10.7717/peerj.17215#supplemental-information.

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
