# Peer review of "Assessing the quality and reliability of YouTube videos as a source of information on inflammatory back pain"

_PeerJ, doi:10.7717/peerj.17215_

## Round 0.1 · original submission · Major Revisions

Highlight the role of YouTube, its merits, and demerits in clinical sciences.
The methodology section should be updated in detail along with the study design and statistical test and provide proper references.

Clearly describe the significance of the study for future research and applications.

**Language Note:** PeerJ staff have identified that the English language needs to be improved. When you prepare your next revision, please either (i) have a colleague who is proficient in English and familiar with the subject matter review your manuscript, or (ii) contact a professional editing service to review your manuscript. PeerJ can provide language editing services - you can contact us at [email protected] for pricing (be sure to provide your manuscript number and title). – PeerJ Staff

·

Basic reporting

1. Mention the prevalence rate and more details of IBP.
2. Mention in detail the role of you tube, its merits and demerits with references.
3. Mention in detail the need for this review with recent references.
4. Mention the clinical significance of this review to clinicians, researchers, and patients.

Experimental design

5. Include the study design and study setting.
6. Include the character of the videos. (The inclusion criteria are not clear)
7. Mention the MESH keywords for searching the videos.
8. Mention the data extraction procedure in detail.
9. Include the outcome variable measured for the review with its reliability and validity.
10. Mention the statistical analysis part.
11. Mention the statistical software used for the analysis.
12. Mention the kappa score of the reviewers who extracted the data.

Validity of the findings

13. Results are not presented in a clear manner.
14. Mention the acronym of abbreviations when it is used for the first time.
15. The conclusion should be more concise and clear based on the study reports.
16. Mention the future recommendations of the review.

Additional comments

Abstract:
1. Define inflammatory back pain in the background of the study.
2. Include the study design and study setting.
3. Include the character of the videoos.
4. Mention the statistical tests used for the analysis and their criteria.
5. The conclusion should be more concise and clear based on the study reports.
6. Avoid abbreviations in the conclusion.

Reviewer 2 ·

Basic reporting

The study covers the topic of IBP and as a Reviewer I understand the importance of this issue for both patients and medical field.
References are form recent studies which is good.
The study has a well defined and build structure.
Results are reasonable and comparable with other YT topic covering similar topics.

Experimental design

The topic of inflammatory back pain is indeed a part of the huge back pain topic, in my opinion the topic is well defined, group size is decent and the manuscript consists of a ethical approval which is rarely seen in such papers.

Methods are comparable with other aritcles focusing on the topic of Youtube and that also is an advantage.

Validity of the findings

The study shows that YT is not reliable in another medical topic. Statistical analysis is also decent, simple, but shows the key problem presented.

Additional comments

In summary the manuscript shows a study on another YT topic performed in a well known and used fashion by ohter Authors using DISCERN JAMA and GQS scoring systems. The search is done recently, ethical approval was aqcuired, statistiscs is sufficent to cover the topic. The discussion is well thought. Consusions show which videos should be chosen by the viewer to acquire the most reliable knowledge.

---

## Round 0.2 · Minor Revisions

Thank you for revising the manuscript. However, it need further to include strengths of the review and improve the English language. Some sentences are very short and ambiguous. You need to address these issues before publication.

**Language Note:** The Academic Editor has identified that the English language must be improved. PeerJ can provide language editing services - please contact us at [email protected] for pricing (be sure to provide your manuscript number and title). Alternatively, you should make your own arrangements to improve the language quality and provide details in your response letter. – PeerJ Staff

·

Basic reporting

Dear authors,

Thank you for the responses and I am satisfied with the reply.

Now the article is good enough to publish in its current state.

Regards

Experimental design

No comment

Validity of the findings

No comment

Additional comments

Dear authors,

Please add some points related to the strengths of the review.

Regards.

---

## Round 0.3 · accepted · Accept

May be accepted for publication as authors addressed all the comments and revised the manuscript.